# Analysis of the Type 4 Effectome across the Genus *Rickettsia*

**DOI:** 10.3390/ijms232415513

**Published:** 2022-12-08

**Authors:** Joseph A. Aspinwall, Kelly A. Brayton

**Affiliations:** Department of Veterinary Microbiology & Pathology, Washington State University, Pullman, WA 99164, USA

**Keywords:** *Rickettsia*, type 4 secretion system, effector, effectome

## Abstract

*Rickettsia* are obligate intracellular bacteria primarily carried by arthropod hosts. The genus *Rickettsia* contains several vertebrate pathogens vectored by hematophagous arthropods. Despite the potential for disease, our understanding of *Rickettsias* are limited by the difficulties associated with growing and manipulating obligate intracellular bacteria. To aid with this, our lab conducted an analysis of eight genomes and three plasmids from across the genus *Rickettsia*. Using OPT4e, a learning algorithm-based program designed to identify effector proteins secreted by the type 4 secretion system, we generated a putative effectome for the genus. We then consolidated effectors into homolog sets to identify effectors unique to *Rickettsia* with different life strategies or evolutionary histories. We also compared predicted effectors to non-effectors for differences in G+C content and gene splitting. Based on this analysis, we predicted 1571 effectors across the genus, resulting in 604 homolog sets. Each species had unique homolog sets, while 42 were present in all eight species analyzed. Effectors were flagged in association with pathogenic, tick and flea-borne *Rickettsia*. Predicted effectors also varied in G+C content and frequency of gene splitting as compared to non-effectors. Species effector repertoires show signs of expansion, degradation, and horizontal acquisition associated with lifestyle and lineage.

## 1. Introduction

The genus *Rickettsia* is comprised of obligate intracellular, α-proteobacteria from the family *Rickettsiaceae*. Bacteria in this genus colonize arthropods, for which they can be mutualistic, commensal, or pathogenic [1]. Many of their arthropod hosts are themselves parasites to plants and vertebrates. Over evolutionary time, *Rickettsia* species have adapted to take advantage of this exposure to a secondary host, leading to emerging vertebrate and plant diseases caused by *Rickettsia* [2]. The disease-causing *Rickettsia* were the first to be identified and have been the focus of scientific inquiry into the genus due to their pathogenetic nature. However, in recent years, additional *Rickettsia* species have been identified as exclusive endosymbionts to invertebrate hosts. This group includes close relatives of pathogenic *Rickettsia*, as well as more distantly related clades within the genus. This distribution suggests a pattern of pathogenesis gain and loss in the genus, potentially as a strategy for horizontal transmission between parasitic arthropod hosts.

*Rickettsias* have been separated into four groups based on phylogenetic relationship, antigenic profile, and arthropod host. These are the Spotted Fever group, the Typhus group, the Transitional group, and the Ancestral group [3]. The Spotted Fever group is a monophyletic group comprised of bacteria carried by ixodid ticks. These bacteria include pathogens of vertebrates, as well as tick endosymbionts. The monophyletic Transitional group is carried by both insects and mites and can cause disease in vertebrates. The Typhus group is comprised of two closely related species, carried by fleas and lice, which are both capable of causing human disease. Finally, the Ancestral group is a paraphyletic group comprised of *Rickettsia* that do not fit into the previous groups. It contains members that are carried by insects and arachnids. No member of this group has been shown to cause vertebrate disease, however, some members have reacted with antibodies from animals or humans [4,5]. There is an additional group referred to as the Torix group within the genus, which is ancestral to the Ancestral group, however, the lack of sequenced genomes prohibits their use in genomic analysis [1].

As is common for obligate intracellular bacteria, *Rickettsia* species have severely reduced genomes as compared to free living bacteria, with genome size ranging from 2.3 Mb (*R. buchneri*) to 1.1 Mb (*R. typhi*) [6,7]. The obligate intracellular nature of *Rickettsias* decreases opportunities for horizontal gene transfer (HGT), yet genome sequencing has revealed varying levels of genome degradation across the genus [8]. Notably, the more reduced *Rickettsia* genomes belong to the most virulent vertebrate pathogens suggesting the inverse relationship between virulence and genome size in *Rickettsia* [9]. Additionally, the larger genomes tend to have repetitive elements: *R. felis*, *R. bellii*, and *R. buchneri* have genomes with many transposases, *Rickettsia* palindromic elements, and short repeat sequences [10,11,12].

Though the basic host cell infection strategy is well established, little is known about the nuances of the *Rickettsia* interaction with their host cell. *Rickettsia* are capable of secreting effector proteins into the host through multiple secretion pathways including the Rickettsiales *vir* homolog type 4 secretion systems (T4SS) [13]. The T4SS is capable of secreting proteins through host membranes. This makes it ideal for manipulating a host cell during cell entry and for vacuole escape. Currently, fewer than 20 proteins have been identified as secreted effectors in *Rickettsia*. Of these, only 3 have been shown to be secreted through the T4SS [14,15,16]. The slow progress of this characterization is due to the limitation of bacterial culture techniques.

Currently, *Rickettsias* cannot be cultured in cell free media. Because of this, in vitro studies can only be conducted within host cells, and potential for genetic manipulation is severely limited. In addition, the T4SS is necessary for host infection, making T4SS knockout mutants impossible within *Rickettsia*. As a result, any understanding of *Rickettsia*-host interactions relies heavily on bioinformatic analysis and comparison to bacteria that can be grown outside of host cells.

To aid in effector identification, our lab has developed a trainable bioinformatic program to predict T4SS effectors [17,18]. This tool, called Optimal Prediction software for Type 4 effectors (OPT4e), takes advantage of three learning algorithms trained on data sets of known T4 effectors and non-effectors. In the present study, we used OPT4e to predict the T4 effectomes from eight species of *Rickettsia*, representing two species from each of the four phylogenetic groups. We compared the putative effector repertoires, identified homologs, identified a set of core effectors common to all species in the study, and examined domains of interest looking for trends that could explain lifestyle differences. This is the first large scale bioinformatic exploration of T4 effectors in the genus *Rickettsia*, and will provide a foundation for functional analyses of effector proteins.

## 2. Results and Discussion

Two fully sequenced *Rickettsia* genomes were selected from each of the four phylogenetic groups of *Rickettsia* (Figure 1). The Spotted Fever group included *R. rickettsii* str. Sheila Smith and *R. montanensis* str. OSU 85-930; the Typhus group included *R. prowazekii* str. Breinl, and *R. typhi* str. Wilmington, the Transitional group included *R. felis* str. URRWXCal2, *R. akari* str. Hartford, and the Ancestral group included *R. bellii* str. RML369-C and *Rickettsia* sp. MEAM1. Though the Ancestral group is paraphyletic, closely related *Rickettsia* were selected from the group to facilitate comparison with the other groups. The goal of the study was to examine and compare the effectomes of a set of organisms from the genus *Rickettsia* and identify commonalities and distinguishing hallmarks for the subgroups that might lead to a greater understanding of the relationship between host, pathogenicity, evolutionary history and the rickettsal effectome.

To supplement this comparison, available plasmids were analyzed. Three of the eight *Rickettsia* are known to contain plasmids. *R. felis* has two sequenced plasmids, one of which is a potential artifact of genome assembly [19]. As the second plasmid contains no effectors that are not present in the first (Figure 2), it does not have a significant effect on the analysis. *R. bellii* and *R. akari* have also been shown to have plasmids. However, only the plasmid for *R. bellii* is available on NCBI. This plasmid was included even though it is from a different strain of *Rickettsia* than was used for the rest of study.

### 2.1. Effector Prediction Compared to Other Rickettsiales

The deduced proteomes from each genome were analyzed using OPT4e to predict putative effector repertoires. OPT4e predicted a total of 1577 effectors from the proteomes of the selected organisms, or a mean of 197 effectors per genome and a range of 84–345 (Table 1). When compared to other vector-borne organisms in the order such as *Anaplasma marginale* (32), *Anaplasma phagocytophilum* (48), *Ehrlichia muris* (89), and *Ehrlichia ruminantium* (109) only the Typhus group *Rickettsia* (101, 84) have a similar number of predicted effectors to the *Ehrlichia* organisms (Appendix A) [18]. This trend was mirrored in by S4TE, an alternative T4 effector prediction program. Fewer effectors are predicted in *Anaplasma* species, while slightly more are predicted in *Ehrlichia*, and the most in *Rickettsia* (Appendix A). While this pattern may be the result of OPT4e identifying motifs associated with host manipulation within the known effector list, and identifying effectors that are not secreted through the T4SS, different T4SS specific effector prediction programs produce the same trend, if not the same effector set. Moreover, there are distinctions between the two families that explain the large effectome in *Rickettsia* such as the rate of gene loss and the probability of lateral gene transfer for *Rickettsiaceae* as compared to *Anaplasmataceae*. The first difference between the groups is the capacity for conjugation and presence of plasmids in some of the *Rickettsiaceae*, which allows HGT between disparate strains and species within the family. This in turn could help in reducing gene loss and degradation associated with the obligate intracellular life cycle as compared to other Rickettsiales. Plasmids have been identified in members of every group of *Rickettsia* except for the Typhus group. Not surprisingly, this group also has the smallest genome, fewest coding sequences, and fewest predicted effectors of any of the *Rickettsias*. The two members of this group are the only *Rickettsias* with similar predicted effector counts to *Ehrlichia* spp. in the *Anaplasmatacae*. Secondly, analysis of *R. bellii* has revealed that ancestors of *Rickettsia* likely infected amoeba, allowing the common ancestor to acquire effectors from pathogens coinfecting the monocellular eukaryotes [12]. RalF, one of the three known T4 effectors in *Rickettsia* are most closely related to a T4 effector in *Legionella pneumophila*, a bacterial pathogen often found in free living amoeba [15]. Finally, many *Rickettsia* genomes are inundated with repetitive elements and mobile genetic elements [10,11,12]. These factors may have led to an increased abundance of effectors in the common ancestor of *Rickettsia* compared to that of *Anaplasma*, as well as an increase in the ability to maintain genes in the accessory genome and to duplicate and recombine sequences in proximity to repetitive elements and mobile genetic elements.

### 2.2. Analysis of Sets of Homologous Effectors across Rickettsia 

The predicted effectors were compared to the list of homologs (Appendix A) generated from a BLASTp comparison of all coding sequences across the eight genomes resulting in 615 effector homolog sets (EHSs) which includes predicted effectors and homologous proteins even if they were not predicted by OPT4e. Eleven EHSs were removed from consideration as T4 effector sets (Appendix A) based on their domain predictions, for a final list of 604 EHSs. Domains were predicted for all protein sequences in EHSs using the Pfam database [20] resulting in 458 domain predictions across 341 EHSs (Appendix A). After removing EHSs with domains associated with alternate secretion pathways, 449 domains remained across 330 EHSs. Consistent with other effector work, many EHSs contain predicted ankyrin repeats (49), tetracopeptide repeats (22), and leucine rich repeats (11) [21,22]. Although they can be found in other proteins across the tree of life, these motifs are overrepresented in known effectors, and perform essential functions in host manipulation [23,24,25]. Multiple transposase domains, phage-associated domains, and toxin-antitoxin domains are also present in predicted proteins. Proteins with toxin-antitoxin and phage-associated domains have been identified as secreted effectors [26,27].

There are 93 predicted domains, flagged across 81 EHSs, in which >95% of representative sequences are from eukaryotes. Of the original 449 predicted domains, 98 are eukaryotic-like domains (ELDs) from the Effective Database [28]. These domains were found to be associated with colonization of eukaryotic cells in a large-scale analysis of pathogenic bacteria. These two groups contain domains likely associated with host manipulation. Three EHSs from these groups have predicted actin binding domains. Two of them, RickA and Sca2, are known to be associated with actin-based motility [29], the third (RT4EHS_191) is a hypothetical protein only present in *R. bellii*, and *R. rickettsii*. In addition to actin-based motility, obligate intracellular bacteria manipulate actin during internalization, to support the bacteria-containing vacuole, to alter vesicular trafficking and pathogen dissemination [30]. Six EHSs were identified that contain Golgi associated and vesicle trafficking domain. Two of these (RalF and Risk1) are known effectors [15,16]. Since the Golgi is the sorting center of the cell, it is a common target in disease, and can result in altered glycosylation, and cellular trafficking [31]. Twelve EHSs can be linked to the eukaryotic cell cycle, containing predicted domains associated with the anaphase-promoting complex, kinetochore, and microtubule binding, or even DNA condensation and centromere attachment. Six EHSs have apoptosis-associated domains, and five EHSs have domains associated with arthropods. Identified domain associations are listed in the notes of Appendix A.

### 2.3. Gene Fragment Analysis

As discussed above, rickettsial organisms have small and degraded genomes as compared to free living bacteria. This degradation leads to fragmentation and potentially pseudogenization of genes. Gene truncation, or fragmentation, does not necessarily preclude functionality, but the gene may no longer be functionally equivalent. To identify cases in which genes are homologous to portions of larger genes, we searched for instances in which smaller genes met all homolog criteria with a larger gene except the length criteria. In these cases, the smaller gene was designated as a gene fragment (Appendix A). When the homolog sets for all genes in the eight *Rickettsia* genomes are analyzed for gene fragments, 1.33% to 21.44% of annotated coding sequences in the genome are flagged as fragments of larger genes (Table 2). This is a drastic variation between species, and may be a result of different algorithms being used when annotating the genomes. However, when all homolog sets smaller than 300 bp are excluded from the analysis, the variation is reduced to 0.9–9.6%. Interestingly, this correction also decreases the proportion of gene fragments in *R. akari* and *R. rickettsii* by 11.67% and 13.15%, respectively. The majority of analyzed species have between 6% and 10% of annotated coding sequences (larger than 300 bp) flagged as gene fragments (Table 2). The Typhus group, which also has the smallest genomes, has the smallest proportion of gene fragments. This could be a result of conservative CDS annotation. This is supported by the low proportion of gene fragments, even when CDSs smaller than 300 bp are included. It may also be associated with the small genome sizes in this group. The shrinking genomes likely lost gene fragments and pseudogenes more quickly than other *Rickettsia* species.

Effectors have a higher proportion of gene fragments than do other coding sequences. This is likely due to the nature of effectors. These genes are often part of the accessory genome which implies more rapid mutation and the potential for degradation. This change is the most evident in *R.* sp. MEAM1, where 25% of predicted effectors are gene fragments while only 7% of non-effectors from homolog sets that meet the size criteria are gene fragments.

### 2.4. G+C Content Analysis

Along with a higher rate of fragmentation, predicted effectors have a significantly different G+C content than the rest of the genome, with a mean consistently below that of other coding sequences (Figure 3). A similar variation in G+C content has been identified in T4 effectors of *Legionella* [32]. Regions of varying G+C content are often associated with HGT, and the low G+C content seen in *Legionella* effectors has been attributed to HGT, however, the lower G+C content of effectors in *Rickettsia* seems unlikely to be associated with HGT as *Rickettsia* already have very low G+C content and regions of other genomes with even lower G+C content would be uncommon.

### 2.5. Effector Categorization

EHSs are separated into five groups (Figure 4 and Figure 5, Appendix A). The Core Group consists of EHSs with at least one protein in every genome. Group Specific Effectors are only found in the two analyzed genomes from any of the four *Rickettsia* groups. Species Specific Effectors are only present in one species. Plasmid Specific effectors are only present on the analyzed plasmids. The Other group contains all EHSs that do not match any of the first four groups. Of the 604 EHSs, 42 are Core, 50 are Group Specific, 352 are Species Specific, 24 are Plasmid Specific, and 136 are Other. The EHSs in the Other group are split into three categories (GL1, GL2 and GL3) (Appendix A). GL1 EHSs are present in species such that the pattern could be explained by a single gene gain or loss event across the phylogeny. GL2 contains gene patterns that are the result of at least two gene gain or loss events. GL3 contains EHSs showing a minimum of three gene gain or loss evens (Appendix A). A few possibilities of these patterns are illustrated in Figure 5B. GL1 contains 22 EHSs, GL2 contains 98, and GL3 contains 15. Though GL1 contains few EHSs compared to GL2, the EHSs used in this observation excluded Species Specific and Group Specific effectors, all of which could be explained by a single gene gain event.

#### 2.5.1. Core Effectors

Of the 42 Core EHSs, 28 contain sequences that are predicted by OPT4e in every species. Although 14 of the predictions are not consistent across all homologs, we think it unlikely that effectors conserved across all species would be secreted using different systems. Seventeen core EHSs contain domains found primarily in eukaryotes or flagged as ELDs. Two proteins that stand out in this list are Sca4 and Patatin 1 (RT4EHS_043, RT4EHS_051). These two proteins are known effectors in *Rickettsia* with no identified secretion system [33,34]. In addition to being maintained on the *Rickettsia* chromosomes, Patatin 1 is present on all 3 plasmids analyzed, although just over half of the homologs of Patatin1 are predicted by OPT4e. Several core homologs contain domains associated with eukaryotic cytoskeleton interaction, and host replication (RT4EHS_002, RT4EHS_013, RT4EHS_014, RT4EHS_016, RT4EHS_043, RT4EHS_061), which could play a role in bacterial motility or vesicle trafficking. Two of the core proteins predicted are DNA polymerase components. Though these cannot be eliminated out of hand, to our knowledge, there are no known cases of DNA polymerase subunits being secreted by bacteria, making it unlikely that these conserved proteins moonlight as effectors in *Rickettsia*.

#### 2.5.2. Group Specific Effectors

Group specific effectors comprise 52 of the EHSs. Most of them are found in the Ancestral group. As indicated by the name, this group is ancestral to the other *Rickettsia*, the high number of group specific effectors is therefore not surprising. In contrast the Typhus group has only 3 group specific effectors, the lowest number of any of the groups, despite being less closely related to the Spotted Fever and Transitional groups than they are to each other. *R. typhi* and *R. prowazekii* have the most reduced genomes of any of the *Rickettsia*, which may be reflected in the low number of group specific effectors.

#### 2.5.3. Species Specific Effectors

The Species Specific EHSs show the same trends as the overall OPT4e predictions. *R. bellii* has the highest number, followed by *R. felis*. The Typhus group has the fewest, and the other *Rickettsia* fall somewhere in between (Figure 4). In *R. bellii* and *R. felis*, species-specific effectors also comprise 45.6%, and 44.6% of their total predicted effectors, respectively. This is higher than any of the other species. *R.* sp. MEAM1 comes the closest with 29.6% of effectors being species-specific. All other species have putative effectomes comprised of less than 18% species specific effectors.

#### 2.5.4. Plasmid-Borne Effectors

Of the 26 EHSs found on the 3 plasmids analyzed, Patatin1 (RT4EHS_051) and RT4EHS_17 are maintained in the genomes of at least one species of *Rickettsia* and RT4EHS_594 is closely related to chromosomal EHSs. These proteins are addressed in either the core effectors section or host specific section. Of the remaining 23, 8 have predicted domains, all but one of which are associated with DNA binding, or antibiotic resistance. That one exception is an ankyrin repeat containing protein (RT4EHS_587) and was the only plasmid exclusive EHS containing either ELDs or eukaryotic domains.

#### 2.5.5. Other Effectors

The 136 EHSs classified as “Other”, are uniquely capable of illuminating trends within the genus. This group contains proteins that were lost in evolutionary lineages or gained within a single clade, Genes that were lost or gained multiple times, or genes that were potentially spread through HGT between *Rickettsia*. The separation of these EHSs into the 3 groups described above is intended to indicate patterns of gene loss and gain. GL3 is the most likely to have occurred by HGT between different clades of *Rickettsia*, GL2 by either HGT between *Rickettsia* or through differential selection, and GL1 being instances of gene loss or gain at a single node in the evolutionary tree (Figure 5).

Of the 22 GL1 EHSs, seven were found in all members of the Spotted Fever group and Transitional group. One, three, and four EHSs were absent only in the Spotted Fever, Ancestral, and Typhus group, respectively and eight were absent in only a single species (Figure 6A). *R. bellii* and *R. felis* have the highest number of GL2 effectors (63 in both). Interestingly, they share 43 of these, 20 of which are only present in these two species (Figure 6B). In comparison, only 31 EHSs are shared exclusively between *R. bellii* and its group member *R.* sp. MEAM, and only 7 EHSs are exclusive to *R. felis* and *R. akari* (Figure 4). In the GL3 effectors, *R. bellii*, *R. felis*, and *R. montanensis* each contain 13 EHSs, most of which they have in common with each other (Figure 6C). The large number of genes shared between *R. bellii* and *R. felis* has been noted before [3]. It is, however, interesting to note that this trend is maintained, and even exaggerated, in predicted effectors.

### 2.6. Effectors in Pathogenc and Non-Pathogec Rickettsia

One effector was identified in all *Rickettsia* known to be vertebrate pathogens. RT4EHS_031 is a 241–319 amino acid hypothetical protein found in *R. felis*, *akari*, *rickettsii*, *prowazekii*, and *typhi*. HMMer analysis predicts a SieB domain in this protein, which is associated with phage superinfection exclusion. Effector, RT4EHS_196, is found only in the Transitional group and *R. Rickettsia*. This protein is annotated as hypothetical and has no recognized domains (Appendix A).

There are no effector homologs present in all non-pathogenic *Rickettsia* species. However, seven effectors are shared between *R. bellii* and *R. montanensis*. One of these genes is a gene fragment of a larger gene in *R.* sp. MEAM1 (RT4EHS_146, RT4EHS_416), the only other non-pathogenic *Rickettsia*. HMMer predictions for this EHS included multiple ELDs as well as a eukaryote specific domain found in vesicle trafficking proteins (SNAP) (Appendix A).

### 2.7. Arthropod Host Associated Effectors

Some effectors are associated with colonization of specific arthropod hosts. Two effectors (RT4EHS_129, RT4EHS_163) were found only in tick-borne *Rickettsia* (Appendix A). RT4EHS_129 has multiple eukaryotic domains predicted, including a clavanin domain, which can be associated with innate immune system modulation, as well as a domain associated with exocytosis, and another associated with histone modification. In addition, transcription of this gene is upregulated at low temperatures, indicating a possible association with the arthropod host [35]. RT4EHS_163 has a Fic domain which was first identified in bacterial toxins, members of the group have been identified as effectors [36]. An additional four EHSs were shared by all tick specific *Rickettsias* as well as *R. felis*. Though the primary vector for *R. felis* is the cat flea (*Ctenocephalides felis*), it has been detected in wild ticks, and shown to be capable of vertical transmission in at least one tick species [37,38]. Of these, two EHSs have eukaryotic domains. One contains both a helix turn helix domain (HTH3) and an A+T rich interaction domain (ARID) found in eukaryotic transcription factors, while the other contains a nucleotide diphosphate cleavage domain (NUDIX) and a repeat domain (WD40_alt) that is specific to eukaryotes.

No putative effectors were conserved across the genomes of flea-borne *Rickettsias* in this homolog analysis. However, the two putative effectors identified on the plasmid of *Rickettsia* felis (RT4EHS_017, RT4EHS_594) could be associated with colonization of the flea host. Interestingly, along with the low identity match between RT4EHS_594 and RT4EHS_018, a third, much shorter gene in *R.* sp. MEAM1 (RT4EHS_371) has identity to the two genes. Even though it is classified as a gene fragment, RT4EHS_371 matches to both RT4EHS_594 and RT4EHS_018, and is predicted to be an effector. Although not homologous by our definition, these three genes contain a region conserved in all insect-borne *Rickettsia*. The presence of both genes on a plasmid in *Rickettsia felis* also highlights the potential for host switching mediated by HGT within *Rickettsia*.

### 2.8. The Relationship between R. bellii and R. felis

Although *R. bellii* and *R. felis* belong to different clades in the phylogeny of the *Rickettsias* they have the highest numbers of predicted effectors at 296 and 251 effectors included in the final set of EHSs, respectively. Compared to *R. bellii* and *R. felis*, *R.* sp. MEAM1, the other member of the Ancestral group, and *R. akari*, from the Transitional group, have much smaller numbers of putative effectors, suggesting that *R.* belli and *R. felis* may have undergone a genome expansion that included expansion of the effector repertoire. These two organisms also have a notably higher proportion of mobile genetic elements and repeat sequences compared to other *Rickettsia* species, which may correspond to genomic rearrangement leading to a decrease in synteny between these organisms and their closely related *Rickettsia* [3]. This could have facilitated expansion of the effectome through gene duplication and shuffling. However, even with signs of genomic rearrangement, these bacteria do not have a higher proportion of gene fragments than the other analyzed species. In *R. bellii*, 9.46% of genes longer than 300 bp are categorized as gene fragments, while in *R. felis* 13.20% fall into this category (Table 2). These values are lower than all other *Rickettsia* except for the Typhus group, discussed above. The remarkable number of effectors in the “Other” category shared by these two species also suggest either convergent evolution or HGT between these organisms, potentially associated with host adaptation (Figure 6B,C).

Even though *R. bellii* is a tick-borne *Rickettsia*, and the primary arthropod host for *R. felis* is the cat flea, these two *Rickettsia* have a larger potential host pool than the other species in this study. *R. bellii* has been identified in more than 25 species of hard and soft ticks [39], while *R. felis* has been found in, and can persist in hard ticks, fleas, lice, and mosquitos [40,41]. Their large and partially convergent effector repertoires may be an important factor in this broad arthropod host range.

## 3. Materials and Methods

### 3.1. Sequences

Eight fully sequenced *Rickettsia* genomes were selected which are available at NCBI. Genomes used included: *R. bellii* str. RML369-C (CP000087.1), *R.* sp. MEAM1 (CP016305.1), *R. felis* str. URRWXCal2 (CP000053.1), *R. akari* str. Hartford (CP000847.1), *R. rickettsii* str. Sheila Smith (CP000848.1), *R. montanensis* str. OSU 85-930 (CP003340.1), *R. prowazekii* str. Breinl (CP004889.1), and *R. typhi* str. Wilmington (AE017197.1). It became evident that in *R.* sp. MEAM1 many genes annotated as pseudogenes were nearly full-length reading frames with homologs in the other species. Therefore, we generated coding sequences for *R.* sp. MEAM1 pseudogenes. If multiple ORFs were present within the gene, they were assigned the locus tag corresponding to the pseudogene followed by a letter to denote the ORF. All manually curated coding sequences can be found in Appendix A). We included plasmid sequences in our analyses when available. The *R. bellii* isolate AN04 (CP015011.1) and *R. felis* URRWXCal2 (CP000054.1, CP000055.1) were used.

### 3.2. Phylogenetic Analysis

All phylogenetic analysis was conducted using MEGAX [42]. Protein sequences for FtsZ, GltA, GroEL, and rpoB were acquired from the eight *Rickettsia* genomes, and *A. marginale* str. St. Maries, which was used as the out group. Sequences were concatenated and aligned using MUSCLE alignment under base parameters. The start sequences of all concatenated genes were examined, and any alignment overlap from one protein sequence to another was manually corrected. The phylogenetic tree was constructed using the JTT maximum likelihood model with 10,000 replicates.

### 3.3. Effector Prediction

We used the OPT4e program to predict effectors from the deduced proteomes. Three sequences of known type 4 secreted effectors from *Rickettsia* (RT0135, RT0362, RT0600) to the default training set [14,15,16]. Though homologs to these proteins are found in multiple species, only the protein sequences from *R. typhi* were used. Protein sequences with internal stop codons cannot be run through this program, so sequences with internal stop codons were run as the open reading frame before the first internal stop codon. Outputs were then used for further analysis. In addition to the above, OPT4e was also used to predict effectors in *A. marginale* str. St. Maries (CP000030.1), *E. ruminantium* str. Springbokfontein7 (CP040111.1), and *E. muris* str. AS145 (CP006917.1) using the original training data set without the addition of T4 effectors from *Rickettsia*. We used published OPT4e prediction data for *A. phagocytophilum* from Ashari et al., 2019.

Effector prediction was conducted using S4TE 2.0 [43]. *Anaplasma* and *Ehrlichia* species listed above were accessed via the publicly available data set provided by S4TE 2.0, while *R. bellii*, *R. montanensis*, and *R. typhi* were uploaded as genbank files and analyzed. As *E. ruminantium* str. Springbokfontein7 was not available, str. Gardel was used in its place.

### 3.4. Homolog Identification

To generate a homolog set across the eight genomes, we used an all versus all BlastP search of all proteins within the genomes to identify regions of high similarity between protein sequences. This search was executed with a local download of Blast+ version 2.10.1 [44]. For protein comparison, we only included blast hits that had an E value of less than 1 × 10^−15^ and a percent identity greater than 40%. If the combined length of all blast hits between two proteins, excluding overlap between blast hits, was greater than 60% of the length of both proteins, these were considered homologs and used in further analysis. If the length of all blast hits was greater than 60% of one protein and less than 60% of the other, the shorter protein was designated as a gene fragment and analyzed separately. All other hits were discarded. All homolog sets in which the longest gene was shorter than 300 bp were removed from the data set. When generating the effector homolog list, all homolog sets in which less than 50% of the homologs were predicted to be secreted through the T4SS were eliminated from the final list.

### 3.5. Protein Domain Identification

To characterize proteins, we used HMMER3.3.1 to check protein sequences against the PfamA database [20,45]. All identified domains above the default threshold for the program were used for further analysis.

### 3.6. Domain of Interest Identification

To identify domains of interest, we compared the list of Pfam domains predicted by HMMer to the data set of Eukaryotic Like Domains (ELDs) available on effectivedb.org [28]. To identify additional eukaryotic domains, we looked at each domain manually on the Pfam website and identified the origin of the sample sequences for the domain on the tree of life. Domains in which >95% of the sequences originated from eukaryotes were designated as predominantly eukaryotic domains.

### 3.7. Comparison of G+C Content

Using the genomes listed above, the G+C content for all genes was calculated and genes were separated into effectors (genes found in the final effector list), and non-effectors (genes that did not fall into the final effector list). These two groups were then evaluated for normality using the Shapiro–Wilks test with an alpha value of 0.05, and the populations were compared by t-test or Mann–Whitney test based on the results.

## 4. Conclusions

OPT4e predicts a large number of T4 effectors across the genus *Rickettsia*. This result is particularly striking when compared to the lower prediction counts of closely related vacuole-bound intracellular bacteria. *R. bellii* and *R. felis*, the largest *Rickettsia* genomes in this study, have the highest number of effectors, the highest proportion of effectors, and the most effectors in common, despite their distance on the phylogenetic tree and difference in lifestyle. Of the 604 predicted EHSs, more than 60% were found in only a single species. The core and group-specific effectors comprise an additional 15%. The remaining 135 EHSs comprise the GL1, GL2, and GL3 groups. The GL1 group shows patterns of gene gain or loss within clades, however, the GL2 and GL3 groups show indications of HGT and convergent evolution. In the GL2 group, with 98 EHSs, 44% are shared between *R. bellii* and *R. felis*, while in the GL3 group, 11–12 of the 15 EHSs are shared between *R. bellii*, *R. felis*, and *R. montanensis*. This indicates a pattern of HGT between the separate groups of *Rickettsia*. Interestingly, two of the three species that stand out in this effector analysis are not known to be vertebrate pathogens. This suggests that these common effectors may be associated with colonization of arthropods rather than vertebrates.

## Figures and Tables

**Figure 1 ijms-23-15513-f001:**
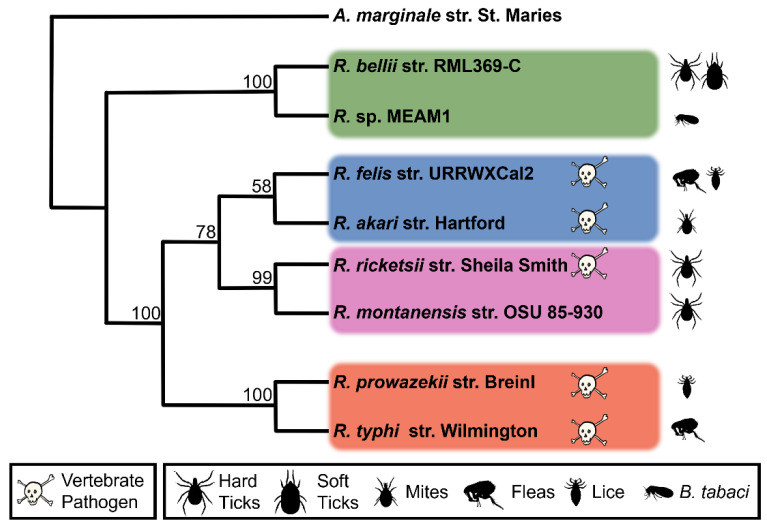
Maximum likelihood phylogenetic tree of the eight species of *Rickettsia* used in this study. *Rickettsia* groups are demarcated by colored rectangles and vertebrate pathogens are marked with a skull and crossbones. Bootstrap values are given at corresponding nodes. *Anaplasma marginale* is used as an outgroup to root the tree. Silhouettes represent known arthropod hosts.

**Figure 2 ijms-23-15513-f002:**
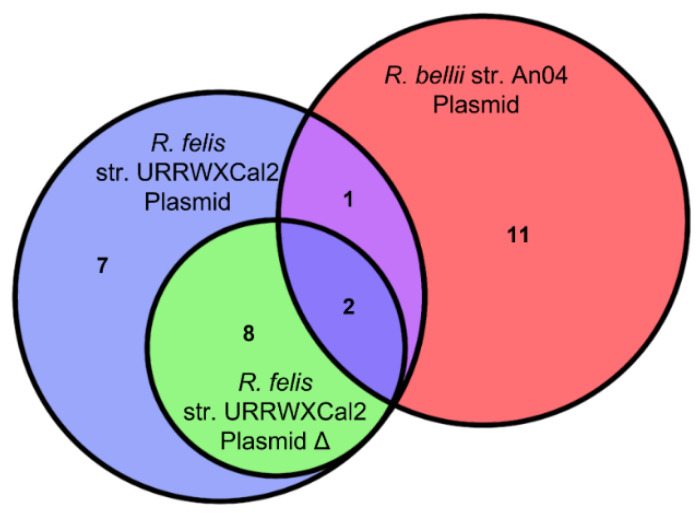
The overlap of predicted effectors between the three analyzed rickettsial plasmids.

**Figure 3 ijms-23-15513-f003:**
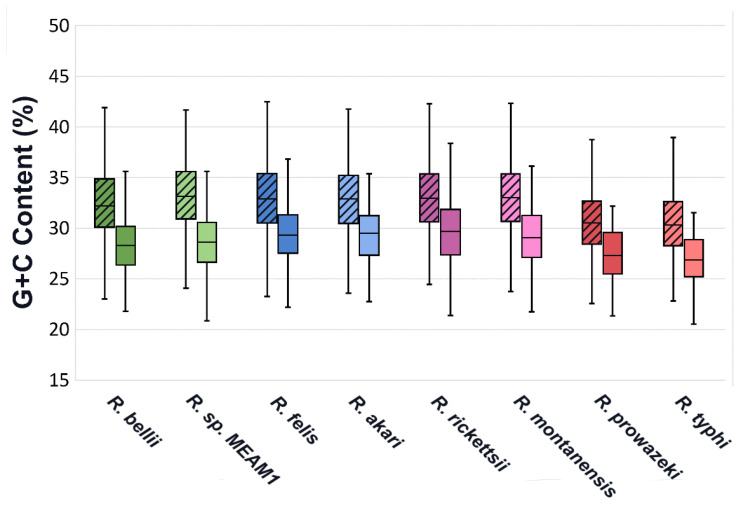
G+C content range of EHSs for each species compared to all other coding sequences. Non-effector genes are represented by plots with diagonal hatching, while Effectors are in open plots. Effector sets are significantly different than Non-effectors in all species, with *p* values below 0.001 in all instances.

**Figure 4 ijms-23-15513-f004:**
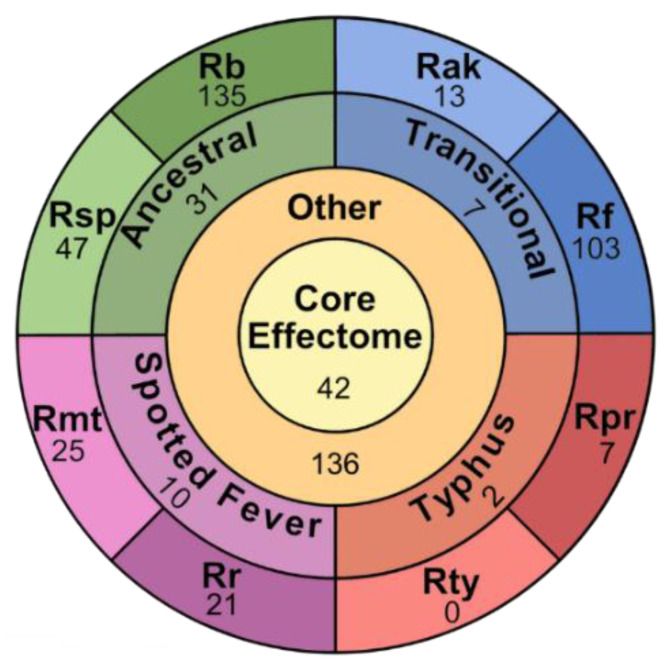
Categorization of 580 chromosomal EHSs. Each section contains a number corresponding to the number of EHSs found in the category. The center circle labeled “core effectome” represents all chromosomal EHSs conserved in all analyzed species of *Rickettsia*. The “other” ring represents the EHSs that are not core, but also not specific to a single group or species. The “group” ring is split into four segments, representing the four groups of *Rickettsia* analyzed in this study. Finally, the “species” ring is split into eight segments, representing each species used in this study. The 2–3 letter labels correspond to the species name with *Rb* standing in for *R. bellii* str. RML396-C, *Rsp* for *R.* sp. MEAM1, *Rf* for *R. felis* str. URRWXCal2, *Rak* for *R. akari* str. Hartford, *Rr* for *R. rickettsia* str. Sheila Smith, *Rmt* for *R. montanensis* str. OSU 85-930, *Rpr* for *R. prowazekii* str. Breinl, and *Rt* for *R. typhi* str. Wilmington.

**Figure 5 ijms-23-15513-f005:**
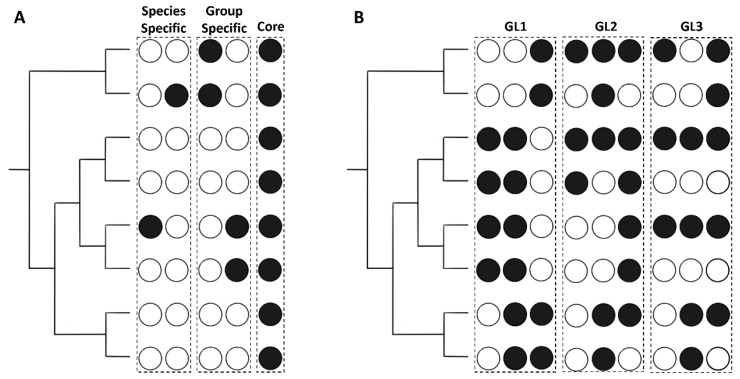
Examples of the categorization scheme used for the EHSs. Circles correspond to the species at the tips of the phylogenies to the right of the figure. Black circles indicate the presence of a homolog in the species, while white circles represent an absence. Instances that meet the conditions for a specific category are grouped within a dashed box and the corresponding group is labeled above. (**A**) shows the instances in which EHSs would be categorized into Species Specific, Group Specific, or Core. (**B**) focuses on the Other group, and gives examples of instances that would be categorized as GL1, GL2, or GL3.

**Figure 6 ijms-23-15513-f006:**
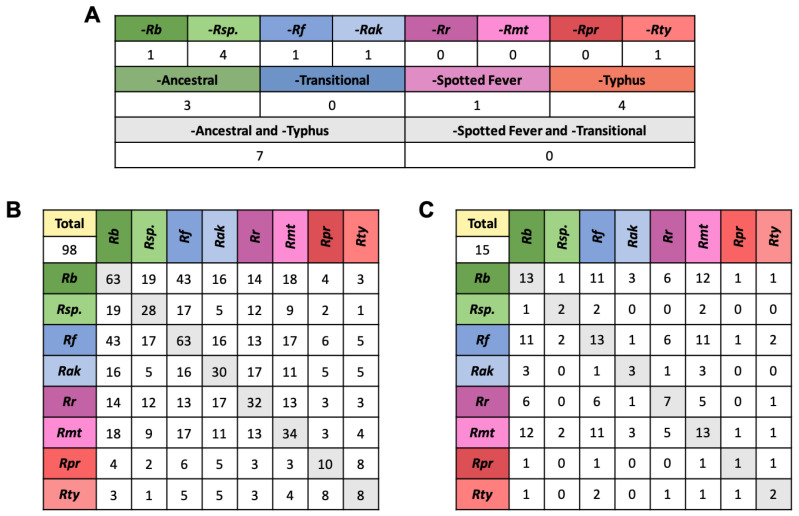
Representation of GL1,2,3 EHSs across *Rickettsia*. (**A**) shows the GL1 genes that are absent in a single species, a single group, or a combination of groups, indicating the evolutionary node at which a gain or loss event likely occurred. (**B**,**C**) show the number of putative effectors from the GL2 and GL3 group, respectively, that are present in every species, as well as how many they share with other species. This shows patterns of shared genes between distantly related species. For an example of how homologs are placed in these tables see Appendix A. The 2–3 letter labels correspond to species with *Rb* standing in for *R. bellii* str. RML396-C, *Rsp.* for *R.* sp. MEAM1 (*B. tabaci*), *Rf* for *R. felis* str. URRWXCal2, *Rak* for *R. akari* str. Hartford, *Rr* for *R. rickettsia* str. Sheila Smith, *Rmt* for *R. montanensis* str. OSU 85-930, *Rpr* for *R. prowazekii* str. Breinl, and *Rt* for *R. typhi* str. Wilmington.

**Table 1 ijms-23-15513-t001:** Predicted Effectors Compared to Coding Genes across the Genus *Rickettsia.*

	*R. bellii*	*R.* sp. MEAM1	*R. felis*	*R. akari*	*R. rickettsii*	*R. montanensis*	*R. prowazekii*	*R. typhi*	Total
Effectors	345 (25%)	203 (14%)	291 (23%)	171 (17%)	186 (15%)	196 (17%)	101 (12%)	84 (10%)	1571 (17%)
Coding Genes	1400	1419 *	1257	1034	1230	1178	839	815	9021

* Gene count in *R.* sp. MEAM1 includes re-annotated pseudogenes.

**Table 2 ijms-23-15513-t002:** Gene Fragment Concentration in Different EHSs.

	*R. bellii*		*R.* sp. MEAM1		*R. felis*		*R. akari*		*R. rickettsii*		*R. montanensis*		*R. prowazekii*		*R. typhi*	
	Gene Frg	Total	Gene Frg	Total	Gene Frg	Total	Gene Frg	Total	Gene Frg	Total	Gene Frg	Total	Gene Frg	Total	Gene Frg	Total
EHSs	28 (9%)	296	39 (25%)	158	33 (13%)	250	18 (15%)	117	26 (19%)	134	28 (19%)	144	9 (12%)	75	1 (2%)	66
Non-EHSs	47 (5%)	897	52 (7%)	786	58 (7%)	891	43 (6%)	745	49 (6%)	770	45 (6%)	791	19 (3%)	701	6 (1%)	685
>300 bp	75 (6%)	1193	91 (10%)	944	91 (8%)	1141	61 (7%)	862	75 (8%)	904	73 (8%)	935	28 (4%)	776	7 (1%)	751
<300 bp	33 (25%)	134	68 (41%)	165	43 (29%)	151	170 (46%)	370	210 (49%)	425	96 (38%)	252	25 (19%)	131	4 (5%)	73

## Data Availability

Publicly available data were analyzed in this study. All data were downloaded from NCBI (www.ncbi.nlm.nih.gov/genome) on 14 May 2020. Specific accession numbers are listed Section 3.1.

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
