# Peer review of "Analysis of the Type 4 Effectome across the Genus Rickettsia"

_ijms, 2022, doi:10.3390/ijms232415513_

Round 1
Reviewer 1 Report
The paper is very well written and presented. Even that is a similar paper like other published about T4SS predicted effectors it is of interest in Rickettsia field. I would like a bit more explanation on the eukaryotic domains found.
It a similar paper like the same group has publishes about the prediction of effectors in Legionella, Coxiella and Anaplasma. The paper is well explained, and the figures are easy to visualise and understand. The only interesting and attractive part is that they have chosen a pathogen difficult to work on laboratory conditions. The paper will contribute to the growing list of T4SS effectors that will need to be tested in vivo.
Author Response
We thank the reviewer for their comments.
In response to their request for a "bit more explanation on the eukaryotic domains found.", we have added a paragraph (in red) to the manuscript in section 2.2 on homologous sets of effectors. To develop the new paragraph, a couple sentences were deleted from the preceding paragraph, so I am pasting both below. In addition, we have added 3 references to the paper to support the new paragraph.
The predicted effectors were compared to the list of homologs (Table S3) generated from a BLASTp comparison of all coding sequences across the eight genomes resulting in 615 effector homolog sets (EHSs) which includes predicted effectors and homologous proteins even if they were not predicted by OPT4e. Eleven EHSs were removed from consideration as T4 effector sets (Table S4) based on their domain predictions, for a final list of 604 EHSs. Domains were predicted for all protein sequences in EHSs using the Pfam database [21] resulting in 458 domain predictions across 341 EHSs (Table S4, Table S5). After removing EHSs with domains associated with alternate secretion pathways, 449 domains remained across 330 EHSs. Consistent with other effector work, many EHSs contain predicted ankyrin repeats (49), tetracopeptide repeats (22), and leucine rich repeats (11) [22, 23]. Although they can be found in other proteins across the tree of life, these motifs are overrepresented in known effectors, and perform essential functions in host manipulation [24, 25, 26]. Multiple transposase domains, phage-associated domains, and toxin-antitoxin domains are also present in predicted proteins. Proteins with toxin-antitoxin and phage-associated domains have been identified as secreted effectors [28, 29].
There are 93 predicted domains, flagged across 81 EHSs, in which >95% of representative sequences are from eukaryotes. Of the original 449 predicted domains, 98 are eukaryotic-like domains (ELDs) from the Effective Database [27]. These domains were found to be associated with colonization of eukaryotic cells in a large-scale analysis of pathogenic bacteria. These two groups contain domains likely associated with host manipulation. Three EHSs from these groups have predicted actin binding domains. Two of them, RickA and Sca2, are known to be associated with actin-based motility [30], the third (RT4EHS_191) is a hypothetical protein only present in R. bellii, and R. rickettsii. In addition to actin-based motility, obligate intracellular bacteria manipulate actin during internalization, to support the bacteria-containing vacuole, to alter vesicular trafficking and pathogen dissemination [31]. Six EHSs were identified that contain Golgi associated and vesicle trafficking domain. Two of these (RalF and Risk1) are known effectors [16, 17]. Since the Golgi is the sorting center of the cell, it is a common target in disease, and can result in altered glycosylation, and cellular trafficking [32]. Twelve EHSs can be linked to the eukaryotic cell cycle, containing predicted domains associated with the anaphase-promoting complex, kinetochore, and microtubule binding, or even DNA condensation and centromere attachment. Six EHSs have apoptosis-associated domains, and five EHSs have domains associated with arthropods. Identified domain associations are listed in the notes of Table S4.
Reviewer 2 Report
Interesting article that analyse the genus Rickettsia throught a bioinformatic exploration. The article is well written and provides data of interest. All the sections are well developed. The bibliography is relevant. The graphs and tables add value to the work.
The only comment is that it would be necessary to detail to which corresponds the skull in Figure 1.

Author Response
We thank the reviewer for their positive response.
The only comment is that it would be necessary to detail to which corresponds the skull in Figure 1.
The legend to the figure indicates "vertebrate pathogens are marked with a skull and crossbones". It is not completely clear to me exactly what information the reviewer is looking for, however, in an effort to be more clear, we added the skull to the pictorial legend in the figure as attached. If that does not address the reviewer's request, we are happy to add more detail if they can be more explicit as to what details they are looking for.